# Th2 Cytokines Affect the Innate Immune Barrier without Impairing the Physical Barrier in a 3D Model of Normal Human Skin

**DOI:** 10.3390/jcm12051941

**Published:** 2023-03-01

**Authors:** Elena Donetti, Federica Riva, Serena Indino, Giulia Lombardo, Franz Baruffaldi Preis, Elia Rosi, Francesca Prignano

**Affiliations:** 1Department of Biomedical Sciences for Health, Università degli Studi di Milano, 20133 Milan, Italy; 2Histology and Embryology Unit, Department of Public Health, Experimental and Forensic Medicine, University of Pavia, 27100 Pavia, Italy; 3Plastic Surgery Unit, Ospedale Niguarda Ca’ Granda, 20162 Milan, Italy; 4Section of Dermatology, Department of Health Sciences, Università degli Studi di Firenze, 50125 Florence, Italy

**Keywords:** human epidermis, immunofluorescence, transmission electron microscopy, keratinocytes, interleukins, toll-like receptors, human beta-defensin 2, involucrin, filaggrin, claudin-1

## Abstract

(1) Background: Atopic dermatitis is one of the most common inflammatory skin diseases characterized by T helper (Th) 2 and Th22 cells producing interleukin (IL)-4/IL-13 and IL-22, respectively. The specific contribution of each cytokine to the impairment of the physical and the immune barrier via Toll-like receptors (TLRs) is poorly addressed concerning the epidermal compartment of the skin. (2) Methods: The effect of IL-4, IL-13, IL-22, and the master cytokine IL-23 is evaluated in a 3D model of normal human skin biopsies (n = 7) at the air–liquid interface for 24 and 48 h. We investigated by immunofluorescence the expressions of (i) claudin-1, zonula occludens (ZO)-1 filaggrin, involucrin for the physical barrier and (ii) TLR2, 4, 7, 9, human beta-defensin 2 (hBD-2) for the immune barrier. (3) Results: Th2 cytokines induce spongiosis and fail in impairing tight junction composition, while IL-22 reduces and IL-23 induces claudin-1 expression. IL-4 and IL-13 affect the TLR-mediated barrier largely than IL-22 and IL-23. IL-4 early inhibits hBD-2 expression, while IL-22 and IL-23 induce its distribution. (4) Conclusions: This experimental approach looks to the pathogenesis of AD through molecular epidermal proteins rather than cytokines only and paves the way for tailored patient therapy.

## 1. Introduction

Atopic dermatitis (AD) is one of the most common inflammatory skin diseases associated with a wide burden and poor quality of life [1,2]. Even if AD is dominated by type T helper (Th) 2 cells and type 2 innate lymphoid cells, other immunopathogenetic pathways seem to play a role according to the many clinical phenotypes [3]. The onset of AD as well as the most acute clinical forms, are associated with high amounts of pro-inflammatory cytokines such as interleukin (IL)-4, IL-13 released by Th2, and IL-22 released by Th17 and Th22 [4]. IL-23 is a core cytokine in many chronic inflammatory disorders, and the IL-17/23 axis is crucial in the pathogenesis of psoriasis [5]. If IL-23 has a specific role in supporting autoimmunity in peripheral tissues, it is not fully validated yet, but it can stabilize the Th17 phenotype and keep their survival [6]. Dendritic cells and macrophages of lesional AD produce high amounts of IL-23, suggesting its role in the initiation and maintenance of skin inflammation [7].

Keratinocytes (KCs), the most represented cytotype in the human epidermis, actively participate in both the physical/chemical barrier and the immunological shield. KCs proliferate in the basal layer and, in the suprabasal layers, undergo a finely tuned and dynamic morpho-functional rearrangement of the cytoskeleton and intercellular junctions, defined as terminal differentiation. A switch occurs from keratin (K) K5/K14 in the basal compartment to K1/K10 in the suprabasal differentiating layers [8]. Inducible keratins K16 and K17 are associated, respectively, with keratinocyte activation and wound healing, and keratinocyte proliferation [9,10]. In parallel with the cytoskeletal rearrangement, the key structural protein filaggrin, present in the keratohyalin granules in the granular layer, binds to keratin intermediate filaments promoting the formation of the most differentiated KCs, i.e., corneocytes [11,12,13]. Finally, yet importantly, the late differentiation stage involves the expression of involucrin, providing mechanical strength to corneocytes themselves [14].

Among the different intercellular junctions, tight junctions (TJs), together with desmosomes, represent the first element for the inside-out barrier in the epidermis. Their structure consists of transmembrane integral membrane proteins belonging to the claudin family and scaffold proteins, i.e., zonula occludens family (ZOs), found in the plaque, permitting the connection between TJ transmembrane proteins and cytoskeletal actin filaments [15].

As mentioned above, KCs actively participate as initiators in innate immunity via Toll-like receptors (TLRs) expression and signaling pathways [16]. TLRs belong to the family of pattern recognition receptors (PRRs), which are expressed on both immune cells and non-immune cells [17]. At least ten TLR subtypes have been identified in humans with specific cellular localization and ligands. TLR1, TLR2, TLR4, TLR5, and TLR6 are located on the cell surface. TLR2 is involved in the binding of residues from Gram-positive bacteria, fungi, parasites, and viruses, and TLR4 responds to LPS, a lipopolysaccharide component of the outer membrane of Gram-negative bacteria. TLR7, TLR8, and TLR9 are nucleic acid-sensing TLRs in the endoplasmic reticulum. TLR7 and TLR8 recognize viral single-strand RNA (ssRNA), whereas TLR9 binds unmethylated 2′-deoxyribocytidine-phosphate-guanosine (CpG) DNA motifs frequently present in bacteria and viruses but rare in mammalians [18].

In the normal human epidermis, TLR4 expression is restricted to basal keratinocytes [19,20]. On the other hand, TLR2 and TLR7 are spread throughout the entire epidermal compartment [21,22], while TLR9 can be expressed only occasionally in the granular layer but is often absent [23].

Activation of different TLRs positively regulates the expression of antimicrobial peptides that include, but are not limited to, defensins. The expression of some of these peptides, such as human beta-defensin 1 (hBD-1), is constitutive. In contrast, the expression of others, including human beta-defensin 2 (hBD-2), is triggered by injury or inflammation of the skin [24].

Skin barrier defects have been considered an initial step in developing AD [25], and all components of the barrier can participate in this process. IL-4 and IL-13 are known to be major players [26], but their precise involvement and role are still debated. The inflammatory environment affects the cytoskeletal arrangement, inducing an increase of K16 expression in the suprabasal epidermis [27] and a downregulation of K10 expression [28]. Filaggrin [29,30], loricrin, and involucrin [31] expressions may also be reduced in AD patients. Inappropriate TLR response and AMP expression are associated with autoimmune skin diseases, such as psoriasis and atopic dermatitis (AD) [24,32], which share some immune-mediated steps, but their aetiopathogenesis is different and involves specific pro-inflammatory cytokines [33,34]. The approval of dupilumab, the fully human monoclonal anti-Th2 cytokine, shed light on AD pathogenesis, demonstrating an effect beyond Th2 inhibition [35]. A possible role is thus emerging with regard to IL-22 and IL-23, classically considered “psoriatic” cytokines. Moreover, the functions of IL-4 and IL-13 overlap but are not identical, and the need to elucidate the specific contribution of each cytokine in different processes is not fading in view of identifying more and more precise pharmacological targets.

A 3D model of normal human skin biopsies maintained at the air–liquid interface and standardized in our laboratory [36,37,38,39,40] represents a clear and simple approach to investigating the early keratinocyte response to a specific inflammatory stimulus. The presence of the physiological epithelial–mesenchymal cross-talk between the epidermis and the underlying dermis mimics as closely as possible the physiological condition. As blood and lymphatic vessels are virtually absent, this setting allows the study of the response induced by each cytokine within the epidermal compartment, paying specific attention to keratinocytes.

In the present study, the impact of the proinflammatory cytokines Th2, i.e., IL-4 and IL-13, IL-22, and IL-23, on (i) the immune epidermal barrier, i.e., TLR2, 4, 7, 9, and hBD-2 expression, and (ii) TJ molecular composition—claudin-1 and ZO-1 were evaluated by immunofluorescence using the 3D model of normal human skin biopsies (n = 7). As for TLRs, TLR2 and TLR4 were chosen for the quantitative analysis as they comprise the recognition of both Gram-positive and Gram-negative bacteria, respectively. To better characterize the specific influence exerted by IL-4 and IL-13, we investigated the expression of biomarkers of cell differentiation, i.e., K14, K10, K16, K17, filaggrin, and involucrin in the same experimental setting. IL-22 effects on these biomarkers have been reported previously [34]. Finally, ultrastructural analysis by transmission electron microscopy (TEM) allowed the measurements of the intercellular distance as an index of spongiosis. All the experiments were performed with biopsies obtained from all subjects.

## 2. Materials and Methods

### 2.1. 3D Organotypic Human Skin Culture

Bioptic fragments of normal human skin were obtained after abdominal aesthetic surgery from healthy, non-smoking, 20- to 40-year-old caucasian women (n = 7) after written informed consent, in accordance with the ethical standards of the Institutional Committee on human experimentation and the Helsinki Declaration. Biopsies were reduced with sterile scalpel to fragments 1 × 1 cm and overnight cultured at air–liquid interface in a Transwell system with the dermis immersed in the culture medium and the epidermis facing the air [36,37]. The samples were then exposed to IL-4 (50 ng/mL), IL-13 (50 ng/mL), IL-22 (100 ng/mL), or IL-23 (50 ng/mL) (PeproTech, London, UK) for 24 and 48 h, culturing parallel control groups. All the experiments were performed with biopsies obtained from all subjects. Skin fragments (5 × 5 mm) were immersion-fixed in 4% formalin in PBS 0.1 M, paraffin-embedded, and cut by a rotatory microtome (Bio-Optica, Milan, Italy), obtaining at least 40 serial sections (5 μm thickness) for each sample.

### 2.2. Immunofluorescence Qualitative Analysis

Specimens were routinely processed for fixation, paraffin embedding, and microtome cut. At least two immunofluorescence experiments were carried out for the qualitative analysis of each marker in each sample in the experimental conditions reported in Table 1. Unspecific binding site saturation was always carried out with 10% goat serum in 0.1 M PBS pH 7.4 (30 min at RT). Negative technical control was always considered on each slide, thus omitting the primary antibody. In samples incubated with Th2 cytokines, K10/K16 double immunostaining was performed. For secondary antibodies, either Alexa Fluor 488 goat anti-mouse or Alexa Fluor 488 goat anti-rabbit (ThermoFisher Scientific, Rockford, IL, USA; dilution 1:200, 1 h at RT) were used as secondary antibodies. Nuclei were counterstained with 4′,6′-diamidino-2-phenylindole dihydrochloride (DAPI; Sigma-Aldrich, St. Louis, MI, USA; dilution 1:50,000, 5 min at RT), and slides were finally mounted with Mowiol 4-88 (Sigma-Aldrich, St. Louis, MI, USA).

Immunofluorescence analysis was performed with a laser scanning confocal microscope Nikon A1R, using constant acquisition parameters for all the experimental groups (Nikon, Tokyo, Japan). hBD-2 experiments were evaluated by a Nikon Eclipse 80i microscope (Nikon, Tokyo, Japan).

### 2.3. Immunofluorescence Quantitative Analysis

For K10, K14, K16, K17, TLR2, and TLR4, at least three experiments were carried out (two slides/sample; two sections/slide), and two blind investigators measured the positive area in µm^2^ by ImageJ 1.53 on the whole section and normalized on living epidermis area, excluding the stratum corneum on serial photomicrographs acquired with constant parameters. Results are expressed as mean of the ratio positive area/living epidermal area + 1 SD.

### 2.4. Transmission Electron Microscopy and Morphometric Analysis

Specimens were routinely processed for TEM analysis and examined by Talos 120 electron microscope (ThermoFisher Scientific, Rockford, IL, USA).

The quantitative analysis of intercellular spaces in both the basal and the suprabasal compartments was performed on ultrathin sections by ImageJ 1.53 on at least 10 random fields per sample, and results were expressed as the mean of intercellular distance (µm) + 1 SD.

### 2.5. Statistical Analysis

Statistically significant differences were always obtained via Kruskal–Wallis analysis of variance followed by Dunn’s post-hoc test using Prism 9.0.0 (GraphPad Software, Boston, MA, USA). Differences were considered statistically significant when *p* < 0.05.

## 3. Results

### 3.1. Th2 Cytokines Affect the Epidermal Innate Immune Barrier without Impairing the TJ Composition

In control samples, the membrane-associated expression of claudin-1 (Figure 1A,B) increased from the spinous layer upwards. After Th2 cytokine exposure, claudin-1 immunostaining was always confined in the uppermost epidermis (Figure 1, panels C–F), similar to controls. IL-22 strongly reduced claudin-1 immunopositivity (Figure 1, panels G and H), while IL-23 induced the expression of this TJ protein, particularly in the granular layer (Figure 1, panels I and J).

ZO-1 cytoplasmic expression was detected in the most differentiated epidermal layers in control samples (Figure 2, panels A and B).

After Th2 incubation, ZO-1 expression was weak in all the suprabasal layers at 24 h (Figure 2, panels C and E) and even more after 48 h (Figure 2, panels D and F). Conversely, IL-22 and IL-23 induced an evident upregulation of the ZO-1 cytoplasmic expression at T48 in the lower epidermal layers (Figure 2, panels H and J). However, no effect was detected at T24 (Figure 2, panels G and I).

As expected, control groups always showed a homogeneous cytoplasmic TLR2 distribution in the entire epidermal compartment (Figure 3, panels A and B), while TLR4 expression was restricted to basal keratinocytes (Figure 3C) and, at 48 h, only a slight immunopositivity extended upwards (Figure 3D). Similarly to TLR2, TLR7 immunostaining was always present throughout the epidermis, with both a cytoplasmic and perinuclear localization (Figure 3, panels E and F), while TLR9 expression was never detected (Figure 3, panels G and H).

TLR2 immunostaining was reduced transiently only in IL-13-exposed samples after 24 h (Figure 4C), while it was comparable in all other cytokine-exposed samples at both time points (Figure 4, panels A, B, and D; Figure 5, panels A–D), similarly to controls (see Figure 3, panels A and B).

Quantitative immunofluorescence analysis of the TLR2-positive area indicated that only IL-13 induced a statistically significant decrease (Figure 6A).

TLR4 appeared discontinuously expressed in the basal layer after cytokine incubation at 24 h (Figure 4, panels E and G; Figure 5, panels E and G) and was reduced to a variable extent by the different cytokines (Figure 6B), with a statistically significant difference only for IL-4 and IL-22. However, at 48 h, IL-4 inhibited TLR4 expression even more evidently than at 24 h (Figure 4F and Figure 6B), while TLR4 immunopositivity was partially restored in IL-22 samples (Figure 5F) or even higher than in controls in IL-13 and IL-23 groups (Figure 4H and Figure 5H) spreading towards the lower spinous layer (Figure 5H, white arrows).

TLR7 immunolabelling intensity was always induced after Th2 cytokine incubation (Figure 4, panels I-L; compared to Figure 3, panels E and F), with a clear perinuclear localization. At 24 h, TLR7 expression was reduced by IL-23 (Figure 5K) but not by IL-22 (Figure 5I), while at T48, TLR4 immunopositivity was inhibited by IL-22 (Figure 5J) but not by IL-23 (Figure 5 L). In all samples, a cytoplasmic localization was evident (Figure 5, panels I–K), with the exception of the IL-23 group at 48 h (Figure 5L). TLR9 induction was never detected in IL-4 (Figure 4, panels M and N) and IL-23 samples (Figure 5, panels O and P), while IL-22 induced a faint and temporary TLR9 induction in samples incubated for 24 h (Figure 5M). Only IL-13 triggered an evident upregulation of TLR9 expression in the granular layer starting from 24 h (Figure 4, panels O and P).

In controls, hBD-2 expression was always localized in the keratinocyte cytoplasm of the medium spinous layer (Figure 7, panels A and B). Only IL-4 completely inhibited hBD-2 expression throughout the entire epidermal compartment (Figure 7, panels C and D), while IL-13 had no effect (Figure 7, panels E and F). On the other hand, IL-22 and IL-23 induced hBD-2 expression throughout the suprabasal compartment at 24 h, an event which became more and more evident at 48 h (Figure 7, panels G–J).

### 3.2. Epidermal Homeostasis Is Affected Differently by IL-4 and IL-13

In control groups, immunostainings for filaggrin (Figure 8, panels A and B) and involucrin (Figure 8, panels C and D) were homogeneously distributed in the cytoplasm of granular keratinocytes with a continuous pattern in between adjacent cells. In all cytokine-treated samples, filaggrin immunolabelling was limited to the uppermost region of keratinocyte cytoplasm (Figure 8, panels E, F, I, and J, inserts) and was interrupted after the incubation only with IL-13 for 48 h (Figure 8, panel J, arrowheads). Involucrin distribution in the epidermal compartment was not affected by any cytokine treatment at T24 (Figure 8, panels G and K), while the immunopositivity faded after the exposure to cytokines for 48 h (Figure 8, panels H and L), in particular in IL-4 group.

In all the samples, when present, keratin immunostaining was localized in the cytoplasm of epidermal keratinocytes (Figure 9).

Similarly to controls (Figure 9, panels A and B), K14 immunolabelling was limited to the basal layer. Its intensity decreased in all cytokine-exposed samples (Figure 9, panels E, F, I, and J), except for the group incubated with IL-4 for 48 h, where the immunopositivity spread toward the suprabasal compartment (Figure 9F, arrows). Its intracellular localization was always evident in correspondence with the basal lamina.

K10 and K16 were expressed homogeneously in the suprabasal and the basal layers of all control samples, respectively (Figure 9, panels C and D). The double immunostaining in cytokine-incubated skin revealed a constant decrease of K10-positive area in the lower spinous layer and a time- and cytokine-dependent induction of K16 expression (Figure 9, panels G, H, K, and L), peaking in IL-13 group after 48 h when it extended upwards the upper spinous layer (Figure 9L).

K17 expression was absent in controls and all samples harvested at 24 h (Appendix A). Only scattered K17-positive keratinocytes were observed after 48 h of cytokine exposure (Appendix A).

The quantitative analysis of the different keratin-positive areas is reported in Figure 9M.

### 3.3. Spongiosis Is an Early AD Event Triggered by Both Th2 Cytokines

By TEM, in all samples, no detachment between the dermis and the epidermis occurred, the basal membrane was uninterrupted, and the fine structure of desmosomes was always preserved (Figure 10, arrows).

In all samples harvested at 24 h, keratin bundles were regularly organized, and chromatin appeared finely dispersed in the nuclei of keratinocytes in both the basal (Figure 10, panels A, E, and I) and the suprabasal compartments (Figure 8, panels B, F and J). In cytokine-treated samples harvested at 48 h, chromatin condensation was observed in the nuclei of basal keratinocytes (Figure 10, panels G and K, asterisks). Aggregated keratin filaments were evident in the basal (Figure 10, panels G and K, arrowheads) and the spinous layer (Figure 10, panels H and L, arrowheads). Compared to controls, the cell-to-cell distance was increased in the cytokine-exposed groups (Figure 10, panels E, F, I, and J), with a statistically significant difference for IL-13 in the basal layer at both time points and for both Th2 cytokines in the suprabasal layers at 48 h (Figure 10, panels M and N).

## 4. Discussion

The identification of the main cellular events underlying AD is more and more challenging because of the multifaceted nature of this disease and the tendency to have heterogeneous clinical features according to the age and medical history of each patient. Furthermore, the initial trigger for AD, the progression mechanisms, and the potential pharmacological intervention require proper experimental skin models to obtain novel insights into new pharmacological targets and tools. In particular, as recently highlighted, “there is an unmet need to better understand epidermal barrier regulation, not only as it applies to general skincare, but also to treatment of common skin conditions, from atopic dermatitis to xerosis” [41]. The 3D normal human skin culture standardized in our laboratory represents a good experimental approach and offers the possibility to study the early impairment of the epidermal barrier in the presence of specific inflammatory stimuli [20,36,39,42].

Our results demonstrate that if, on the one hand, Th2 cytokines fail to impair the physical epidermal barrier, they strongly affect the TLR-mediated innate immune barrier to a greater extent than IL-22. Interestingly, the master cytokine IL-23 does not have any relevant effect on TLR expression, with the exception of TLR4 at 48 h. We can thus hypothesize that, in the early phases of AD, the epidermal barrier is simultaneously impaired on the innate side by Th2 cytokines and on the physical side by IL-22, as indicated in Figure 11.

The clinical features observed in AD lesions can be considered the combined result of these two different cytokine-mediated effects. We also demonstrated that although IL-4 and IL-13 are known to share many regulatory mechanisms, their effects are not identical and interfere specifically with some epidermal phenomena. The histo-morphological approach herein presented is pivotal to understanding if the impairment of protein expression is accompanied by a change of the protein epidermal distribution, an issue otherwise neglected with the molecular analysis and/or in in vitro systems.

We report that claudin-1 expression was not affected when either IL-4 or IL-13 alone was added, confirming the experimental evidence obtained in primary human keratinocytes by Yuki et al. [43] and recently in reconstructed human epidermis by Cadau and Coll. [44]. Our study stands in continuation with in vitro studies [45] and Honzke’s study performed in skin equivalents incubated with Th2 cytokines [46], with the evident advantage of displaying a fully developed epidermal barrier, thus allowing a more realistic and direct extrapolation to the clinics. Conversely, claudin-1 expression was inhibited in the presence of IL-22, suggesting that the impaired TJ structure reported in clinics is due not to Th2 cytokines but to IL-22 itself. The reduced expression of ZO-1 in Th2-incubated samples is in agreement with the findings that (i) only a small amount of ZO proteins is required for nucleating claudin strand assembly, (ii) a minimal scaffold at the junction may be sufficient to set up the initial claudin fibrils, and (iii) claudins have the capacity to self-organize [47]. While the role of IL-23 has been widely discussed in the pathogenesis of psoriasis, due to the specific role of dendritic cells [48], less has been reported concerning AD. To the best of our knowledge, the impact of this master cytokine on TJ molecular composition in the epidermis is not elucidated yet. For this reason, the increased expression of both claudin-1 and ZO-1 induced by IL-23 needs further investigation to understand if this effect can relate to mechanisms other than intercellular junctions, in particular for ZO-1.

Filaggrin expression was reduced slightly in the presence of Th2 cytokines, suggesting that this effect is not one of the earliest and constant features occurring during the initial phases of AD lesion formation, but it can represent a later pathogenic event in AD as a clear impairment of its expression is clinically reported in bioptic samples [49]. An early filaggrin expression downregulation was observed in normal human keratinocytes incubated with IL-4 or IL-13 [49] and in engineered skin equivalents [50], but the different settings can account for this discrepancy. On the whole, our evidence is in accordance with the observations that i) FLG gene mutations are not found in all AD patients with a penetration of just 40% [51], and ii) FLG mutation carriers do not always develop AD [52,53]. A similar conclusion can be drawn for involucrin, as its expression was not affected in our experiments, but the reduction of its presence in AD lesions has been reported both at the gene and the protein level [31].

For the immune barrier, Th2 cytokines involved in AD pathogenesis exert a specific and significant tuning in accordance with the literature for TLR2 and TLR4 in bioptic AD skin [21], demonstrating that this shift in TLR expression may be related to a need for enhanced immune surveillance against microbe invasion. Similarly, TLR7 upregulation is reported in peripheral blood monocytes of AD patients [54]. Regarding TLR9 upregulation induced by IL-13, our findings agree with the boost of IL-1α secretion, an interleukin known to be induced in AD patients [55] and observed in HaCaT cells after incubation with strains of *S. aureus* [56]. The relationship between TLR9 and AD was also confirmed by the blockade of the increase of IL-1α levels by the pretreatment with iODNs, a TLR9 antagonist [55].

Considering the “psoriatic” pro-inflammatory cytokines, i.e., IL-22 and IL-23, contrasting results describing the surface TLR pattern distribution in psoriatic plaques are reported [21,57,58,59]. TLR2/TLR4 upregulation in a psoriatic milieu was reported after stimulation of normal human keratinocytes with IFN-γ and TNF-α [60], but i) the different experimental setting, i.e., 2D vs. 3D, and ii) the different inflammatory stimulus can explain this discrepancy. The absence of a persistent modulation of TLR2/TLR4 expression after exposure either to IL-22 or to IL-23 in our experimental conditions can thus be meaningful. In parallel, the limited effect exerted by IL-22 and IL-23 on TLR7/TLR9 expression should be discussed based on recent observations reporting their upregulation in plaque psoriasis biopsies [61] and after the incubation of 3D organotypic cultures of normal human skin with TNF-alpha, IL-17, IL-22, IL-23, which can reproduce the psoriatic plaque milieu [42]. Both evidences suggest that a complete psoriatic microenvironment is required to modulate TLR7 and TLR9 expression and that a single cytokine is not able to induce any significant tuning in the considered experimental conditions.

Discussing hBD-2 immunoreactivity, considered a useful marker to identify some clinical forms of psoriasis requiring differential diagnosis from DA, the interesting result was that IL-4, but not IL-13, early inhibited its epidermal expression, demonstrating the specificity of the downstream effects induced by each Th2 cytokine. In accordance with previous studies [62], both IL-22 and IL-23 strongly induced hBD-2 expression, thus giving a clear explanation for the higher levels of this antimicrobial protein found in psoriasis than in AD [63].

In our experimental conditions, both Th2 cytokines similarly reduced K10 expression, i.e., an epidermal differentiation marker of terminal differentiation in the suprabasal layers, but only IL-4 impairs the homeostasis in the basal layer, as shown by the rise of K14 expression. Interestingly, the keratinocyte proliferation rate is never influenced by any cytokine (unpublished personal observations). These data stand in continuation with the existing evidence obtained in in vitro keratinocytes [64,65] and strongly suggest that the alteration of the epidermal differentiation early occurs also in the presence of Th2 cytokines. Furthermore, the enlargement of intercellular spaces was evident as early as 24 h after Th2 cytokine incubation, in particular in the IL-13 group, suggesting that spongiosis is an initial AD pathogenetic event specifically triggered by Th2 cytokines, as recently reported [44,45,46].

We previously demonstrated that K17 expression, an important and specific psoriatic marker, can be early induced in the same experimental model after the incubation with a mixture of TNF-alpha, IL-17, IL-22, and IL-23 with a pattern distribution in the epidermal compartment very similar to psoriatic skin described in the literature [42]. As expected, in the present study, Th2 cytokines samples did not induce K17 expression, in accordance with the existing literature [10]. In the presence of an AD milieu, the K16-positive epidermal area progressively increased with time, reaching a statistically significant difference compared to the control group at 48 h. This observation stands in continuation with the existing literature [27] and indicates that, once again, the normal epidermis is ready for early and specifically responding to a proinflammatory microenvironment with the expression of inducible keratins. Similarly to the psoriatic microenvironment, in this case, it seems relevant that the stimulus should be prolonged, thus reproducing as strictly as possible the pathological condition in which pro-inflammatory cytokines are involved. Future studies considering the co-presence of IL-4, IL-13, and IL-22 are needed to investigate a potential additive/synergic effect among these cytokines.

The four compartments of the epidermal barrier, i.e., the physical, chemical, immunological, and microbial barrier [66], can be affected by genetic and environmental factors [67]. AD and psoriasis are the main inflammatory skin diseases associated with impaired skin barrier function. The mechanisms related to epidermal barrier dysfunction may be primary and/or secondary.

Indeed, mutations in the filaggrin gene were identified in a subset of patients with AD. On the other hand, considering that not all patients with AD display filaggrin mutations, a combination of primary and secondary barrier defects underlies the disease process. Thus, the inflammation in AD patients, predominantly characterized by a strong and inappropriate Th2 cell activation [68], affects the integrity of the epidermal barrier on multiple levels. Secondary factors adversely affecting the epidermal barrier integrity are predominant in the pathogenesis of psoriasis. Moreover, lipid abnormalities in the stratum corneum were found in both AD and psoriasis [69]. The epithelial barrier function is also crucial for intestinal homeostasis. Indeed, in some subsets of inflammatory bowel disease (IBD), as emerged from the animal model data, barrier dysfunction may be a primary contributor to the disease (a primary defect) and not a consequence of mucosal inflammation [70].

## 5. Conclusions

TJ dysfunction in the presence of a physiological epidermal stratification/barrier is secondary to Th2 inflammatory processes. Future studies evaluating the functional properties of the epidermal barrier and permeability tests are needed to complete this complex tableau. Nevertheless, this experimental approach looks to the pathogenesis of AD through molecular epidermal proteins rather than cytokines only and paves the way for tailored patient therapy.

## Figures and Tables

**Figure 1 jcm-12-01941-f001:**
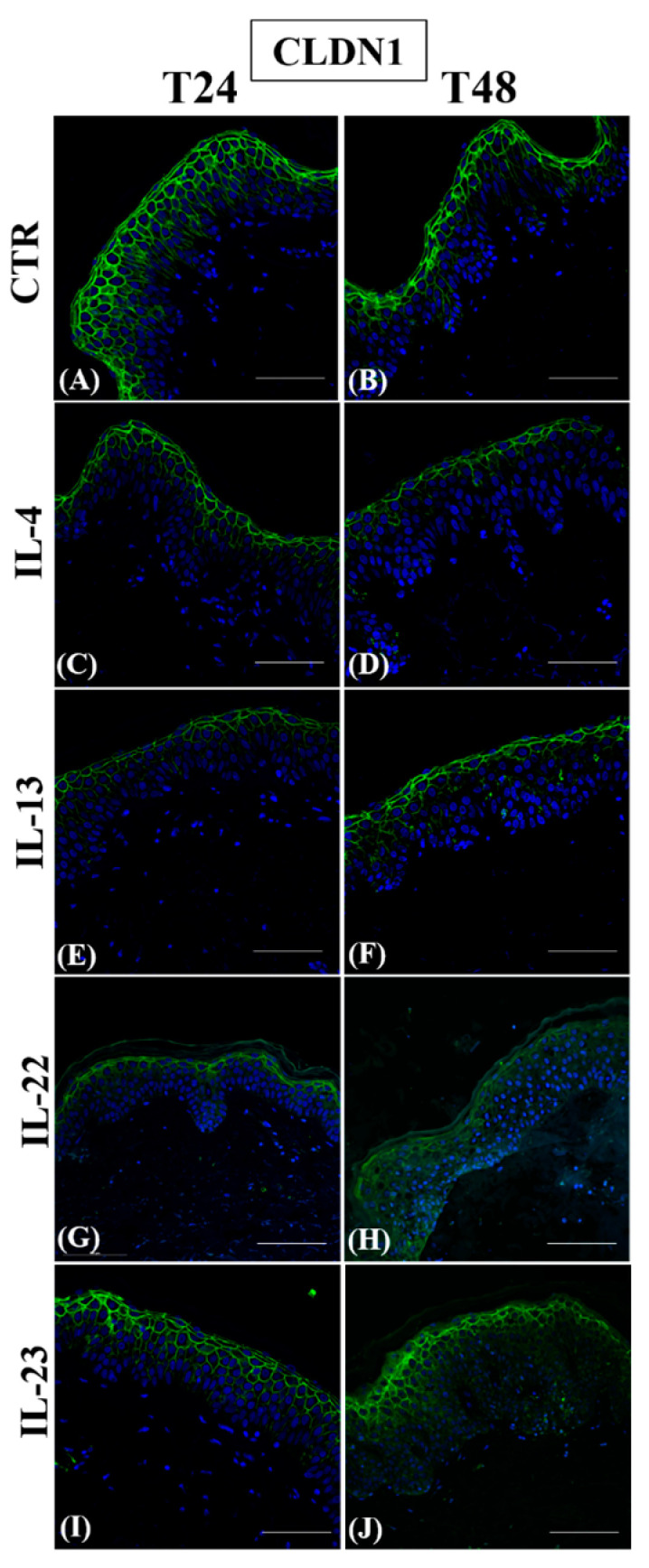
Immunofluorescence of claudin-1 expression on paraffin human skin sections. Representative claudin-1 immunostaining in normal human skin paraffin sections. (**A**,**C**,**E**,**G**,**I**): samples harvested at 24 h; (**B**,**D**,**F**,**H**,**J**): samples harvested at 48 h. (**A**,**B**): CTR samples; (**C**,**D**): IL-4-treated samples; (**E**,**F**): IL-13-treated samples; (**G**,**H**): IL-22-treated samples; (**I**,**J**): IL-23-treated samples. Nuclei are counterstained with DAPI. CTR: control; IL-4: interleukin 4; IL-13: interleukin 13; IL-22: interleukin 22; IL-23: interleukin 23; DAPI: 4′, 6-diamidino-2-phenylindoledihydrochloride—scale bars: 50 µm.

**Figure 2 jcm-12-01941-f002:**
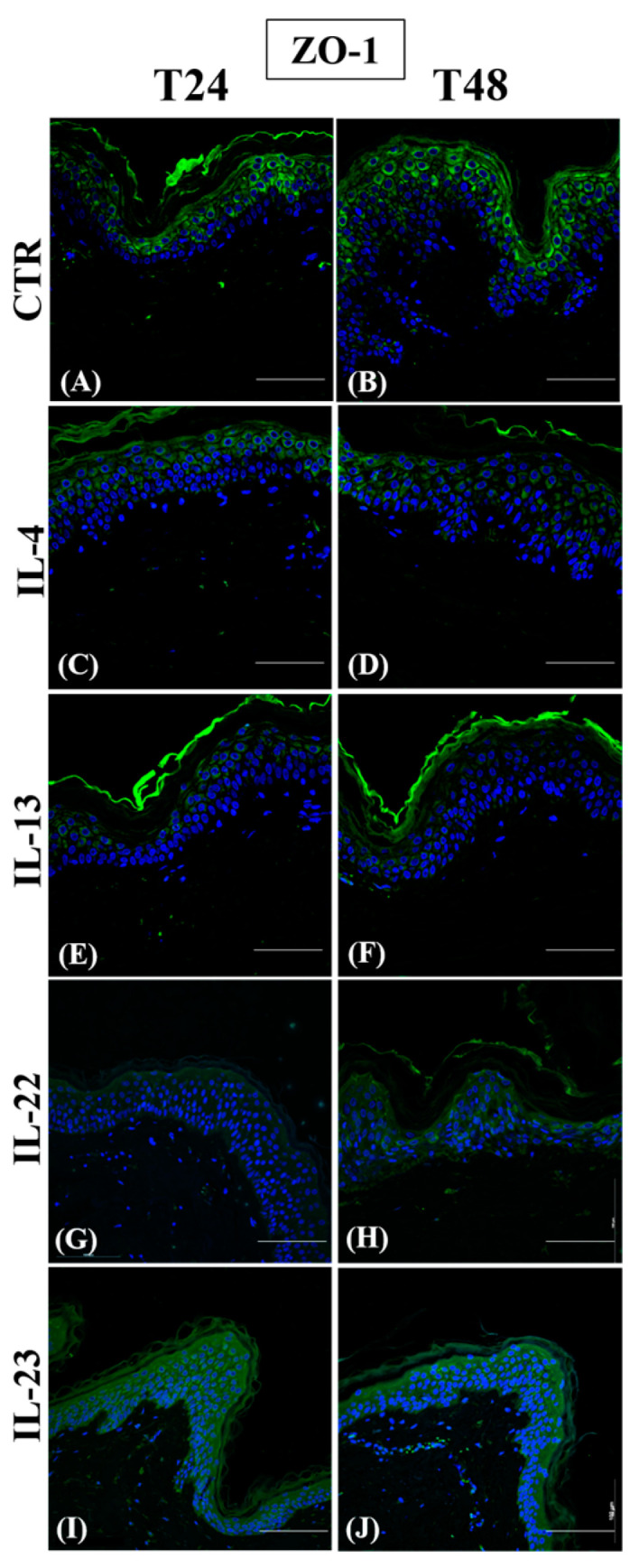
Immunofluorescence of ZO-1 expression on paraffin human skin sections. Representative ZO-1 immunostaining in normal human skin paraffin sections. (**A**,**C**,**E**,**G**,**I**): samples harvested at 24 h; (**B**,**D**,**F**,**H**,**J**): samples harvested at 48 h. (**A**,**B**): CTR samples; (C, D): IL-4-treated samples; (**E**,**F**): IL-13-treated samples; (**G**,**H**): IL-22-treated samples; (**I**,**J**): IL-23-treated samples. Nuclei are counterstained with DAPI. ZO-1: zonula occludens 1. CTR: control; IL-4: interleukin 4; IL-13: interleukin 13; IL-22: interleukin 22; IL-23: interleukin 23; DAPI: 4′, 6-diamidino-2-phenylindoledihydrochloride—scale bars: 50 µm.

**Figure 3 jcm-12-01941-f003:**
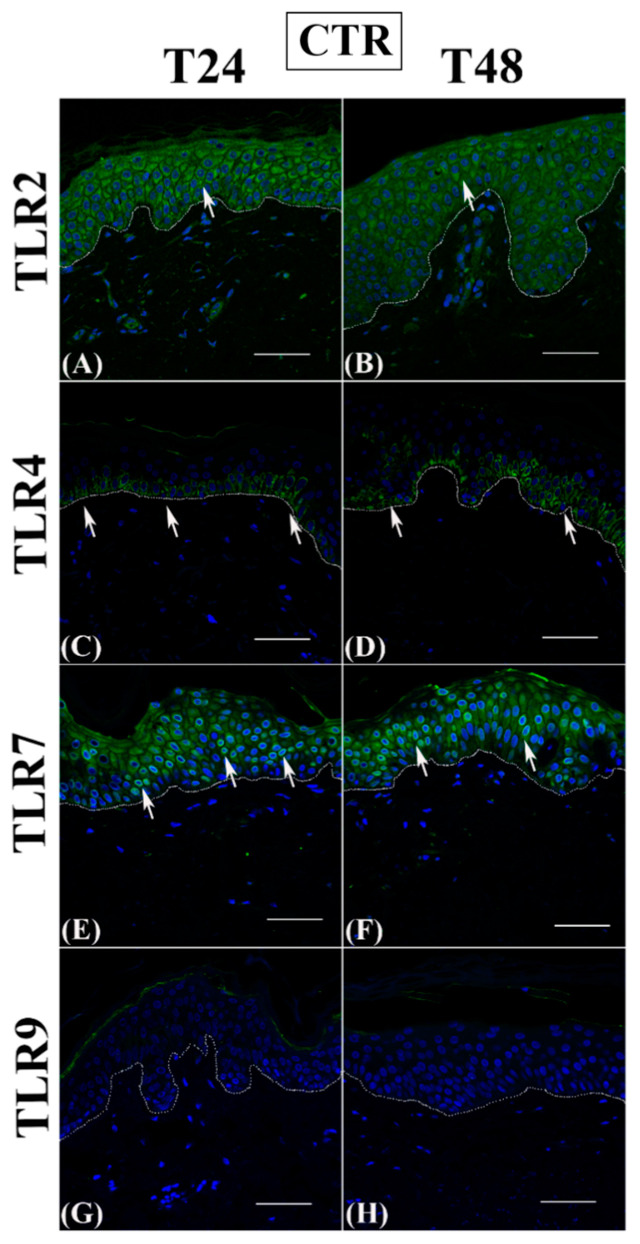
TLR2, TLR4, TLR7, and TLR9 immunofluorescence analysis on paraffin human skin sections. Representative TLR2 (**A**,**B**), TLR4 (**C**,**D**), TLR7 (**E**,**F**), and TLR9 (**G**,**H**) immunostainings in normal human skin paraffin sections. (**A**,**C**,**E**,**G**): samples harvested at 24 h; (**B**,**D**,**F**,**H**): samples harvested at 48 h. (**A**–**H**): control samples. Nuclei are counterstained with DAPI. TLR2: Toll-like receptor 2; TLR4: Toll-like receptor 4; TLR7: Toll-like receptor 7; TLR9: Toll-like receptor 9; DAPI: 4′, 6-diamidino-2-phenylindoledihydrochloride. White dotted line indicates the basal membrane. White arrows indicate positive immunostaining—scale bars: 50 µm.

**Figure 4 jcm-12-01941-f004:**
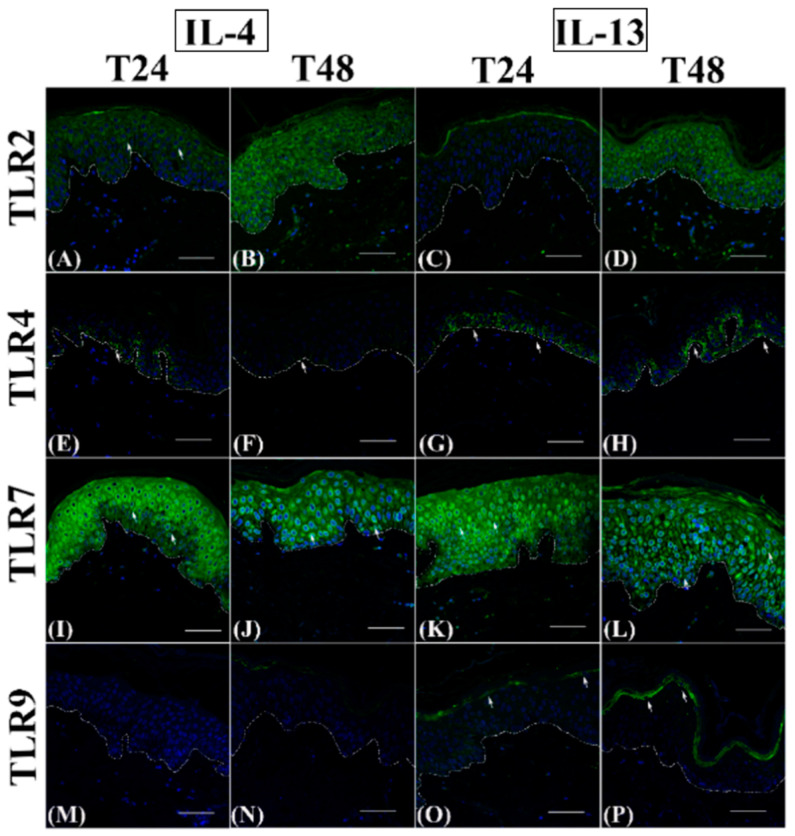
TLR2, TLR4, TLR7, and TLR9 immunofluorescence analysis on IL-4 and IL-13 incubated paraffin human skin sections. Representative TLR2 (**A**–**D**), TLR4 (**E**–**H**), TLR7 (**I**–**L**), and TLR9 (**M**–**P**) immunostainings in normal human skin paraffin sections. (**A**,**E**,**I**,**M**,**C**,**G**,**K**,**O**): samples harvested at 24 h; (**B**,**F**,**J**,**N**,**D**,**H**,**L**,**P**): samples harvested at 48 h. Nuclei are counterstained with DAPI. TLR2: Toll-like receptor 2; TLR4: Toll-like receptor 4; TLR7: Toll-like receptor 7; TLR9: Toll-like receptor 9; IL-4: interleukin 4; IL-13: interleukin 13; DAPI: 4′, 6-diamidino-2-phenylindoledihydrochloride. White dotted line indicates the basal membrane. White arrows indicate positive immunostaining—scale bars: 50 µm.

**Figure 5 jcm-12-01941-f005:**
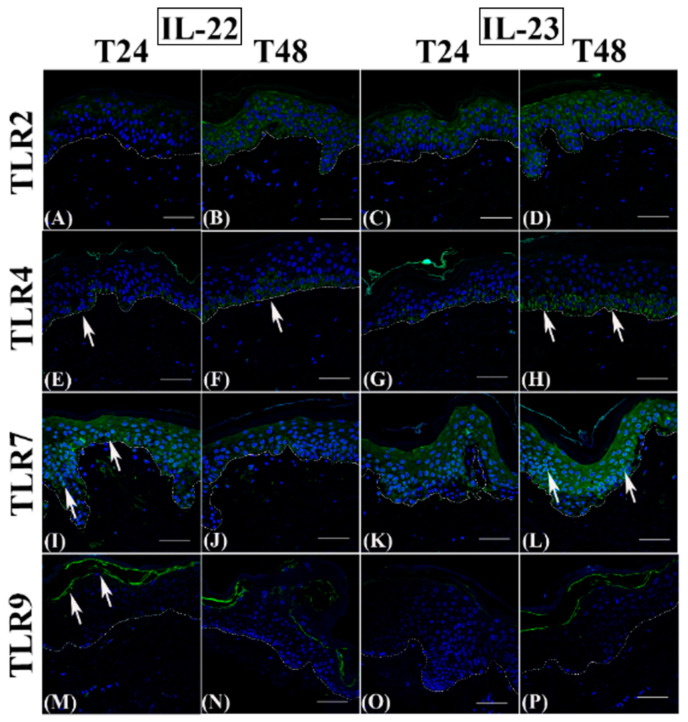
TLR2, TLR4, TLR7, and TLR9 immunofluorescence analysis on IL-22 and IL-23 incubated paraffin human skin sections. Representative TLR2 (**A**–**D**), TLR4 (**E**–**H**), TLR7 (**I**–**L**), and TLR9 (**M**–**P**) immunostainings in normal human skin paraffin sections. (**A**,**E**,**I**,**M**,**C**,**G**,**K**,**O**): samples harvested at 24 h; (**B**,**F**,**J**,**N**,**D**,**H**,**L**,**P**): samples harvested at 48 h. Nuclei are counterstained with DAPI. TLR2: Toll-like receptor 2; TLR4: Toll-like receptor 4; TLR7: Toll-like receptor 7; TLR9: Toll-like receptor 9; IL-22: interleukin 22; IL-23: interleukin 23; DAPI: 4′,6-diamidino-2-phenylindoledihydrochloride. White dotted line indicates the basal membrane. White arrows indicate positive immunostaining—scale bars: 50 µm.

**Figure 6 jcm-12-01941-f006:**
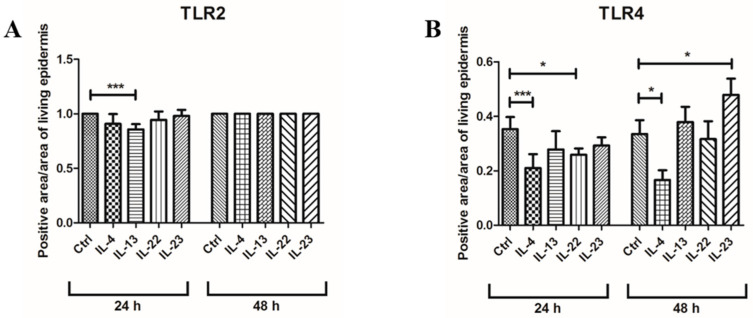
Quantitative analysis of TLR2 and TLR 4 epidermal distribution after immunofluorescence analysis after 24 and 48 h of cytokine incubation. (**A**): TLR2; (**B**): TLR4. TLR2: Toll-like receptor 2; TLR4: Toll-like receptor 4; IL-4: interleukin 4; IL-13: interleukin 13; IL-22: interleukin 22; IL-23: interleukin 23. Statistical analysis was performed by Prism 9.0.0 via Kruskal–Wallis non-parametric analysis of variance, followed by Dunn’s post-hoc multiple comparison test. Differences were considered statistically significant when *p* < 0.05. * *p* < 0.05; *** *p* < 0.01.

**Figure 7 jcm-12-01941-f007:**
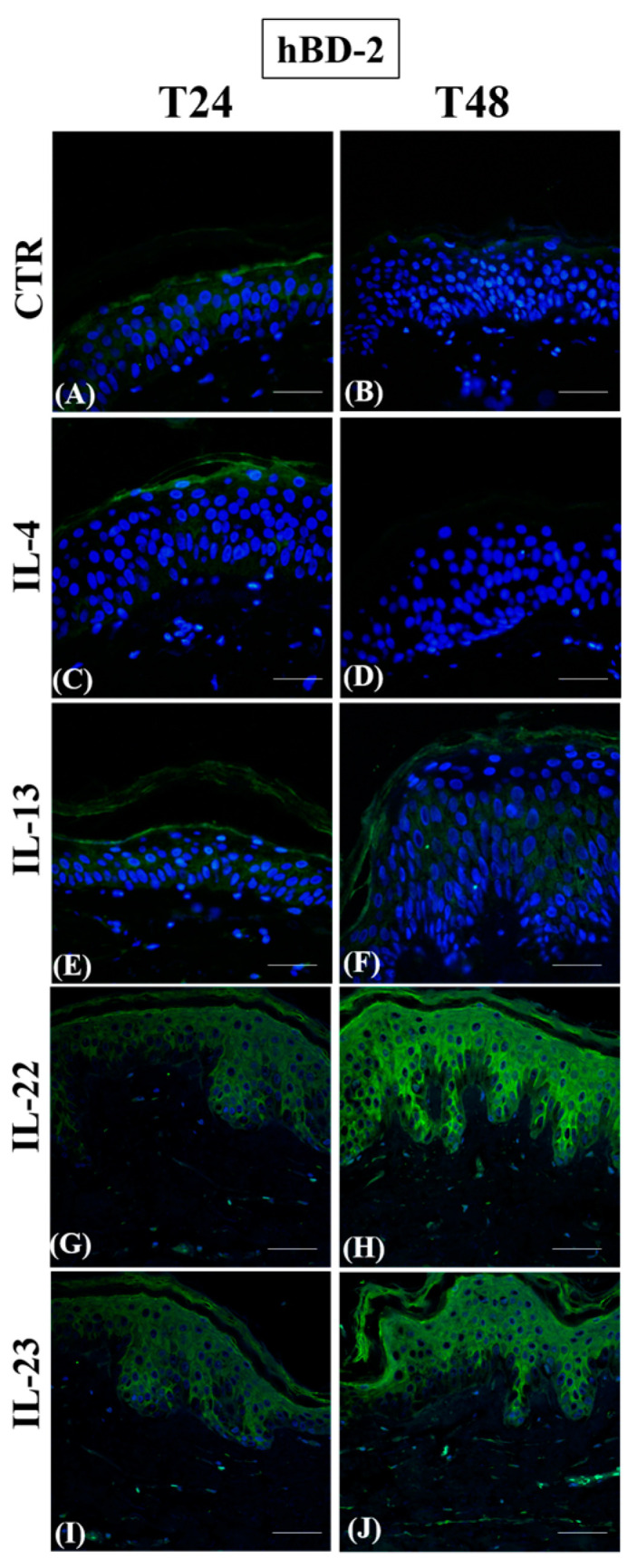
Immunofluorescence of hBD-2 expression on paraffin human skin sections. Representative hBD-2 immunostaining in normal human skin paraffin sections. (**A**,**C**,**E**,**G**,**I**): samples harvested at 24 h; (**B**,**D**,**F**,**H**,**J**): samples harvested at 48 h. (**A**,**B**): CTR samples; (**C**,**D**): IL-4-treated samples; (**E**,**F**): IL-13-treated samples; (**G**,**H**): IL-22-treated samples; (**I**,**J**): IL-23-treated samples. Nuclei are counterstained with DAPI. hBD-2: human beta-defensin 2. CTR: control; IL-4: interleukin 4; IL-13: interleukin 13; IL-22: interleukin 22; IL-23: interleukin 23; DAPI: 4′, 6-diamidino-2-phenylindoledihydrochloride—scale bars: 50 µm.

**Figure 8 jcm-12-01941-f008:**
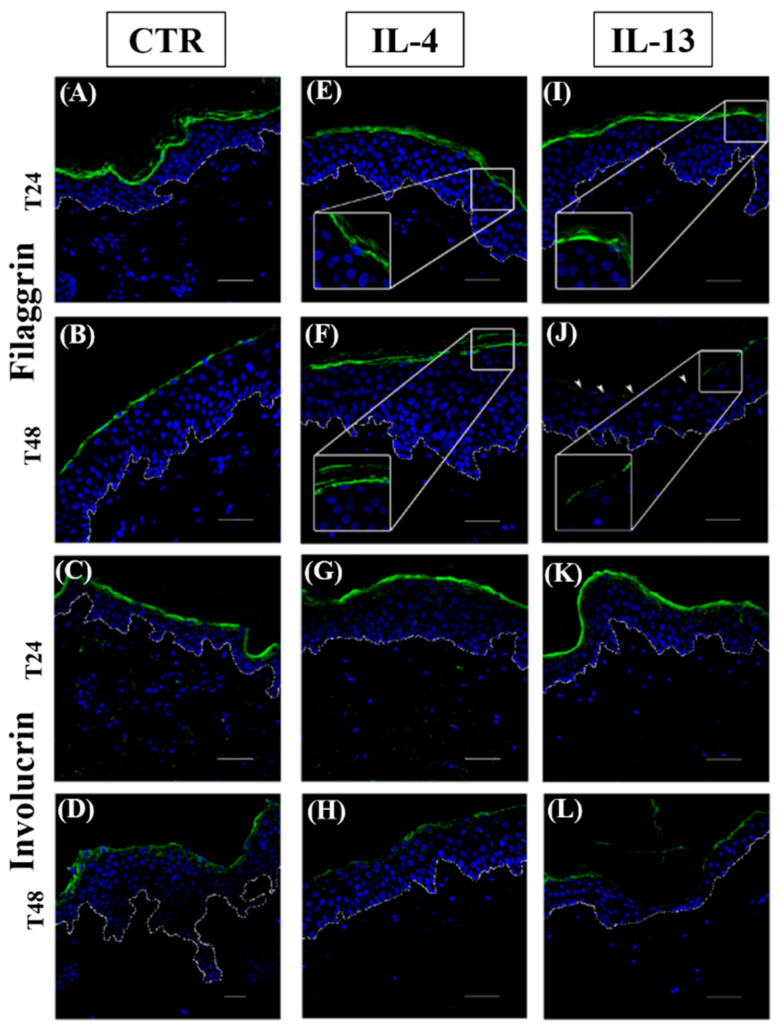
Immunofluorescence analysis of filaggrin and involucrin. Representative filaggrin (**A**,**B**,**E**,**F**,**I**,**J**) and involucrin (**C**,**D**,**G**,**H**,**K**,**L**) immunostainings in normal human skin paraffin sections. (**A**,**C**,**E**,**G**,**I**,**K**): samples harvested at 24 h; (**B**,**D**,**F**,**H**,**J**,**L**): samples harvested at 48 h. (**A**–**D**): CTR samples; (**E**–**H**): IL-4-treated samples; (**I**–**L**): IL-13-treated samples. Nuclei are counterstained with DAPI. CTR: control; IL-4: interleukin 4; IL-13: interleukin 13; DAPI: 4′, 6-diamidino-2-phenylindoledihydrochloride. White dotted line indicates the basal membrane—scale bars: 50 µm.

**Figure 9 jcm-12-01941-f009:**
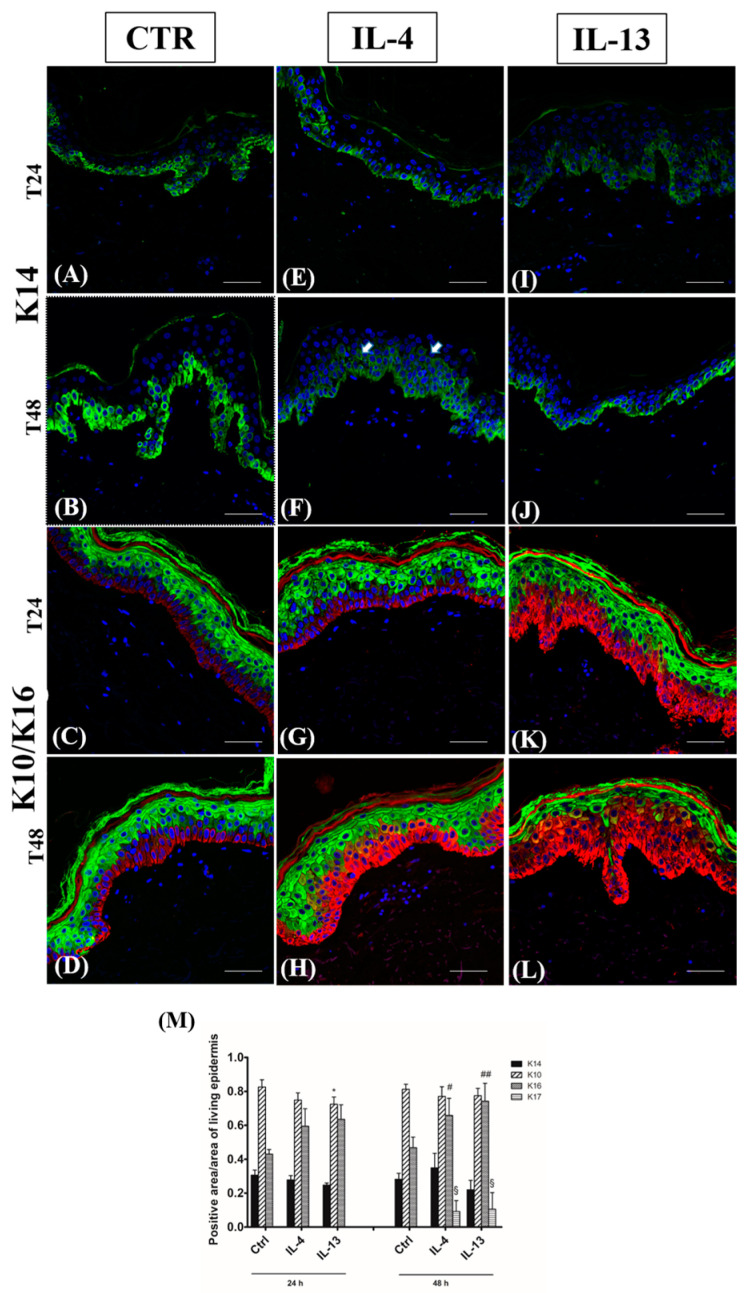
K14 and K10/K16 qualitative and quantitative immunofluorescence analysis. Representative K14 (**A**,**B**,**E**,**F**,**I**,**J**) and K10/K16 (**C**,**D**,**G**,**H**,**K**,**L**) qualitative immunostainings, and (**M**) quantitative analysis in normal human skin paraffin sections. Green staining for K10 and red staining for K16. (**A**,**C**,**E**,**G**,**I**,**K**): samples harvested at 24 h; (**B**,**D**,**F**,**H**,**J**,**L**): samples harvested at 48 h. (**A**–**D**): CTR samples; (**E**–**H**): IL-4-treated samples; (**I**–**L**): IL-13-treated samples. (**M**) Quantitative analysis of K14-, K10-, K16-, and K17-positive areas in normal human skin paraffin sections. Results are expressed as the ratio positive area/area of living epidermis + 1 SD; bars indicate standard error. * *p* < 0.05 vs. all CTR samples; # *p* < 0.01 vs. all CTR samples; ## *p* < 0.005 vs. all CTR samples, § *p* < 0.001 vs. all CTR samples (Kruskal–Wallis analysis of variance followed by Dunn’s post-hoc test). Nuclei are counterstained with DAPI. K14: keratin 14; K10: keratin 10; K16: keratin 16; K17: keratin 17; CTR: control; IL-4: interleukin 4; IL-13: interleukin 13; DAPI: 4′, 6-diamidino-2-phenylindoledihydrochloride. Arrowheads indicate the lower spinous layer. White dotted line indicates the basal membrane—scale bars: 50 µm.

**Figure 10 jcm-12-01941-f010:**
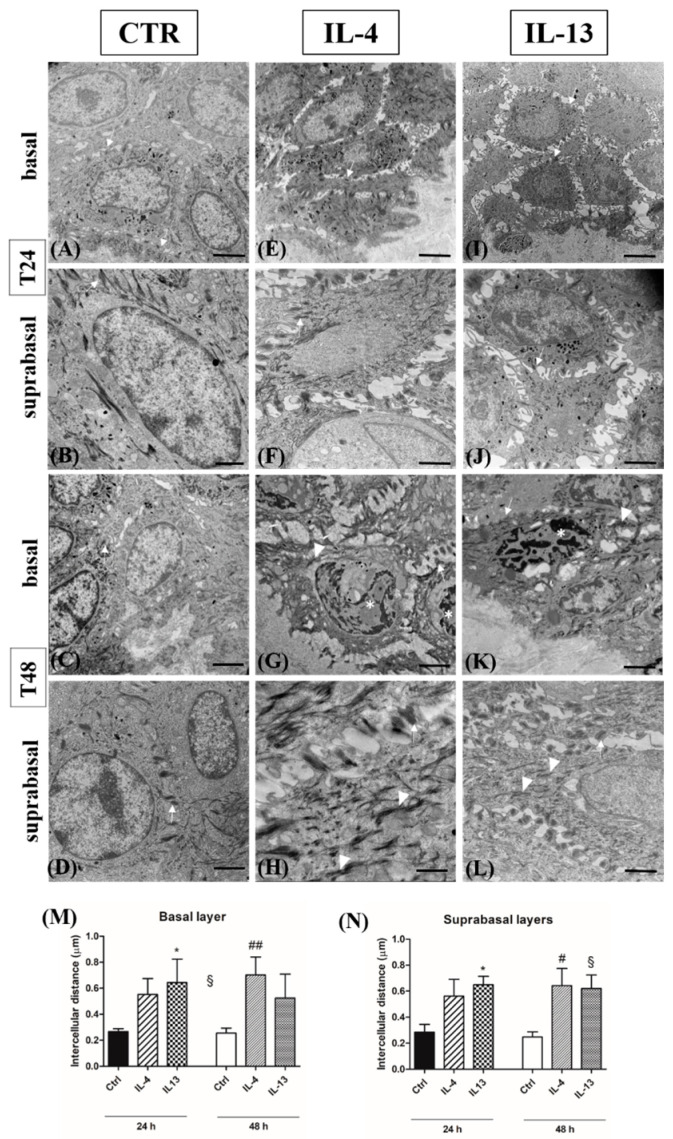
Transmission electron microscopy analysis. Representative photomicrographs of normal human skin araldite ultrathin sections (**A**–**L**) and quantitative analysis of intercellular spaces in the basal (**M**) and the suprabasal layers (**N**). (**A**,**B**,**E**,**F**,**I**,**J**): Samples harvested at 24 h; (**C**,**D**,**G**,**H**,**K**,**L**): samples harvested at 48 h. (**A**,**C**,**E**,**G**,**I**,**K**): Basal layer; (**B**,**D**,**F**,**H**,**J**,**L**): suprabasal layers. (**A**–**D**): CTR samples; (E-H): IL-4-treated samples; (I-L): IL-13-treated samples. Results are expressed as the mean of intercellular distance (µm) + 1 SD; bars indicate standard error. * *p* < 0,05 vs. all CTR samples; # *p* < 0.01 vs. all CTR samples; ## *p* < 0.005 vs. all CTR samples, § *p* < 0.001 vs. all CTR samples (Kruskal–Wallis analysis of variance followed by Dunn’s post-hoc test). CTR: control; IL-4: interleukin 4; IL-13: interleukin 13. Arrows indicate desmosomes; arrowheads indicate keratin filament aggregation; asterisks indicate chromatin condensation—scale bars (**A**,**C**,**E**,**G**,**I**,**K**): 2 µm; scale bars (**B**,**D**,**F**,**H**,**J**,**L**): 1 µm.

**Figure 11 jcm-12-01941-f011:**
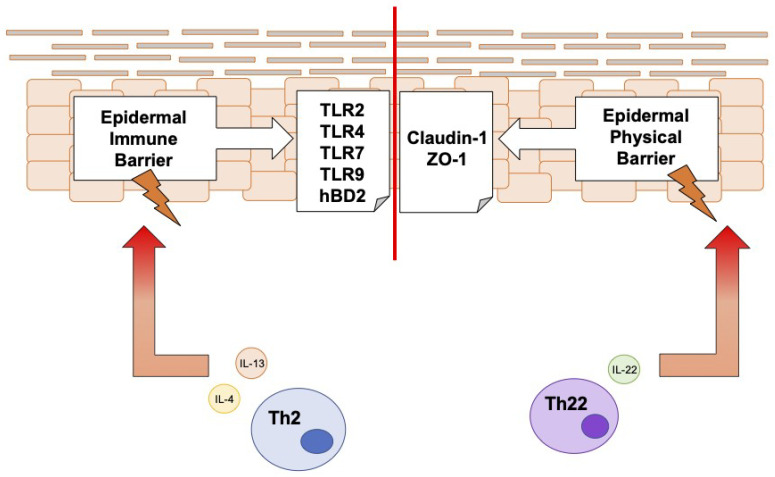
Schematic representation of the early impairment of the epidermal barrier in a proinflammatory microenvironment mimicking atopic dermatitis. TLR: Toll-like receptor; hBD-2: human beta-defensin 2; IL-4: interleukin 4; IL-13: interleukin 13; Th2: T helper (Th) 2 cells; ZO-1: zonula occludens 1; IL-22: interleukin 22; Th22: T helper (Th) 22 cells.

**Table 1 jcm-12-01941-t001:** Antibodies and protocols for indirect immunofluorescence analysis.

Antibody	Antigen Retrieval	Incubation (Antibody Diluted in PBS/BSA 2%)
Polyclonal rabbit anti-human CLDN-1 (ThermoFisher Scientific, Rockford, IL, USA)	0.01 M citrate buffer pH 6 in MW	dilution 1:100 1 h at 37 °C
Polyclonal rabbit anti-human ZO-1 (ThermoFisher Scientific)	Pronase E 10 min at 37°C	dilution 1:100 1 h at 37 °C
Monoclonal mouse anti-human TLR2 (Novus Bio, Littleton, CO, USA)	0.01 M Na citrate buffer pH 6 in MW	1:100 overnight at 4 °C
Monoclonal mouse anti-human TLR4 (Novus Bio)		1:300 1 h at 37 °C
Monoclonal mouse anti-human TLR9 (Novus Bio)		dilution 1:10 overnight at 4 °C
Polyclonal rabbit anti-human TLR7 (Novus Bio)	0.05 M Tris HCl pH 8.5 in MW	1:300 overnight at 4 °C
Polyclonal rabbit-anti-human hBD2 (Santa Cruz Biotechnology, Dallas, TX, USA)	0.01M Na citrate buffer in autoclave 120 °C 6 min	dilution 1:50 overnight 4 °C
Monoclonal mouse anti-human filaggrin (Santa Cruz Biotechnology)	0.01 M Na citrate buffer pH 6 in MW	dilution 1:250, overnight at 4 °C
Monoclonal mouse anti-human involucrin (ThermoFisher Scientific)		dilution 1:1000, overnight at 4 °C
Monoclonal mouse anti-human K10 (Santa Cruz Biotechnology)		dilution 1:50 overnight at 4 °C
Monoclonal rabbit anti-human K16 (Bio SB, Santa Barbara, CA, USA)		dilution 1:100, 1 h at 37 °C
Rabbit anti-human K17 (Abcam, Cambridge, UK)		dilution 1:200, overnight at 4 °C
Monoclonal mouse anti-human K14 (Santa Cruz Biotechnology)	pepsin 0.05% 15′ RT and 0.01M Na citrate buffer pH 6 in MW	1:200 overnight at 4 °C

CLDN-1: claudin-1; ZO-1: zonula occludens 1; TLR: Toll-like receptor; hBD-2: human beta-defensin 2; K: keratin; MW: microwave; RT: Room Temperature; PBS: Phosphate Buffer Saline; BSA: Bovine Serum Albumin.

## Data Availability

In this study no new data were created.

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
