# Peer review of "Th2 Cytokines Affect the Innate Immune Barrier without Impairing the Physical Barrier in a 3D Model of Normal Human Skin"

_jcm, 2023, doi:10.3390/jcm12051941_

Round 1

Reviewer 1 Report

The paper is about differences abotu topic dermatitis and the study of the effect of Th2 cytokines onthe innate barrier and the physical barrier in a 3d model of normal human skin.In particular authors studied the effect of IL-4, IL-13, IL-22 and IL-23 in order to investigate by immunofluorescence the expressions of iclaudin 1, zonula occludens (ZO)-1, filaggrin, involucrin for the physical barrier and TLR2, 4, 7, 9, human Beta Defensin 2 (hBD2) for  the immune barrier. In the studied model Th2 Th2 cytokines were able to significantly affect the immune barrier without impairing, contestually, the physical barrier.

The paper is well written, methods are detailed and well conducted. Also, the statistical analysis is well done.

Author Response

We thank the Reviewer for his/her appreciation of our work.

Reviewer 2 Report

The manuscript titled “Th2 cytokines affect the innate immune barrier without impairing the physical barrier in a 3D model of normal human skin” has some interesting findings but requires some clarifications before the manuscript can be considered for publication.

Authors report the effects of Th2 and Th22 cytokines on mediating innate immunity and their response to the physical barrier in a 3D human skin model. The results show that cytokine effects differed, which is an interesting finding in AD, as cytokines play a pivotal role in AD development.

Some of the questions to be addressed are as below.

  1.    Can authors provide information on the race of the participants from whom the normal human skin was collected?

  2.    Authors mention the number of participants was n=7, but it is unclear if the experiments were performed from skin collected from all seven participants. Please clarify.

  3.    I would encourage the authors to show the supplemental figures as main figures, as the findings from filaggrin, involucrin, and ZO-1 are important observations. Keep the K17 figure as the supplemental figure.  

  4.    The results reported from exposures of IL-22 and IL-23 are on Claudin-1, TLR2, TLR4, TLR7 and TLR9 and hBD2. I cannot find the results for filaggrin and involucrin when the skin is exposed to IL-22 or IL-23. But Figure 9 shows Th22 effects on filaggrin and involucrin. Can authors explain and modify the figure as needed?

  5.    Figure 7 does not show K17. Please remove the wording for K17 from the figure legend.

Author Response

  1. Can authors provide information on the race of the participants from whom the normal human skin was collected?

In the revised version, we specified “caucasian” at line 127.

  1. Authors mention the number of participants was n=7, but it is unclear if the experiments were performed from skin collected from all seven participants. Please clarify.

In the original manuscript the sentence was: All the subjects were represented in all the experimental groups at both the experimental time points (line 136). Now we modified as follows: “All the experiments were performed in biopsies obtained from all subjects” (line 121 and line 134)

  1. I would encourage the authors to show the supplemental figures as main figures, as the findings from filaggrin, involucrin, and ZO-1 are important observations. Keep the K17 figure as the supplemental figure.

We really appreciated the Reviewer’s encouragement and Figure S1 (ZO-1) is now Figure 2 and S2 (filaggrin and involucrin) is now Figure 8 and all the Figures were renamed. Table 1 was updated with all the considered markers. As suggested, in the revised version, only K17 Figure is in the supplementary material as Figure S1.

  1. The results reported from exposures of IL-22 and IL-23 are on Claudin-1, TLR2, TLR4, TLR7 and TLR9 and hBD2. I cannot find the results for filaggrin and involucrin when the skin is exposed to IL-22 or IL-23. But Figure 9 shows Th22 effects on filaggrin and involucrin. Can authors explain and modify the figure as needed?

We totally agree with the Reviewer and we modified as suggested the Figure 9 (now Figure 11).

  1. Figure 7 does not show K17. Please remove the wording for K17 from the figure legend.

Actually, panel M in Figure 7 (now Figure 9) reports the quantitative analysis of the distribution of K10, K14, K16, and K17, which was detected only at T48 in samples incubated with IL-22 and IL-23. For this reason, K17 is mentioned in the figure legend. Moreover, we updated the legend of Figure S1.

Reviewer 3 Report

The authors present an interesting article about ther role of Th2 affecting the innate immune barrier. The article is weel written and interesting. Some minor revisions to do:

- please add a shor paragraph about the main diseases thata re associated with alteration of the immune barrier (e.g. inflammatory bowel diseases, psoriasis, atopic dermatitis)

- please in the discussion speculate also about the impact of Vitamin D on the immune barriere and how this can change also according to the sun exposion. In this regard please read and add these paper (Vitamin D receptor immunohistochemistry variability in sun-exposed and non-sun-exposed melanomas. Melanoma Res. 2017 Feb;27(1):17-23. doi: 10.1097/CMR.0000000000000311. PMID: 27792059.).

Author Response

  1. Please add a shor paragraph about the main diseases thata re associated with alteration of the immune barrier (e.g. inflammatory bowel diseases, psoriasis, atopic dermatitis)

As requested, we added a short paragraph at the end of the discussion at lines 524-540 with the related references (66-70) in the Reference Section as follows

The four compartments of the epidermal barrier, i.e the physical, chemical, immunological and microbial barrier [66] can be affected by both genetic and environmental factors [67]. AD and psoriasis are the main inflammatory skin diseases associated with impaired skin barrier function. The mechanisms related to epidermal barrier dysfunction may be primary and/or secondary.

Indeed, mutations in the filaggrin gene were identified in a subset of patients with AD. On the other hand, considering that not all patients with AD display filaggrin mutations, a combination of primary and secondary barrier defects underlies the disease process. Thus, the inflammation in AD patients, which is predominantly characterized by a strong and inappropriate Th2 cells activation [68], affects the integrity of the epidermal barrier on multiple levels. Secondary factors adversely affecting the epidermal barrier integrity are predominant in the pathogenesis of psoriasis. Moreover, lipids abnormalities in the stratum corneum were found in both AD and psoriasis [69].         
The epithelial barrier function is also crucial for intestinal homeostasis. Indeed, in some subsets of inflammatory bowel disease (IBD), as emerged from the animal model data, barrier dysfunction may be a primary contributor to disease (a primary defect) and not a consequence of mucosal inflammation [70].

  1. Please in the discussion speculate also about the impact of Vitamin D on the immune barriere and how this can change also according to the sun exposion. In this regard please read and add these paper (Vitamin D receptor immunohistochemistry variability in sun-exposed and non-sun-exposed melanomas. Melanoma Res. 2017 Feb;27(1):17-23. doi: 10.1097/CMR.0000000000000311. PMID: 27792059.).

We really appreciate the Reviewer’s suggestion and we red the suggested paper. We are aware that the impact of Vitamin D on the immune epidermal barrier and on keratinocyte differentiation is a key topic, but it is beyond the aim of the present study.